# Novel Strategy in Searching for Natural Compounds with Anti-Aging and Rejuvenating Potential

**DOI:** 10.3390/ijms24098020

**Published:** 2023-04-28

**Authors:** Andrey Koptyug, Yurij Sukhovei, Elena Kostolomova, Irina Unger, Vladimir Kozlov

**Affiliations:** 1SportsTech Research Center, Department of Engineering, Mathematics and Science Education, Mid Sweden University, Akademigatan 1, 831 25 Östersund, Sweden; 2Institute of Fundamental and Clinical Immunology, Tyumen Branch, Kotovskogo Str. 5, 625027 Tyumen, Russia; 3Department of Microbiology, Tyumen State Medical University, Kotovskogo Str. 5/2, 625023 Tyumen, Russia; 4Institute of Fundamental and Clinical Immunology, Department of Clinical Immunology, Yadrintcevskaya Str. 14, 630099 Novosibirsk, Russia

**Keywords:** regeneration, rejuvenation, anti-aging potential, search strategy, proliferation niche, cell culture, mononuclear cells, fibroblasts, cell age profile, cell cycle stages, flow cytometry

## Abstract

We suggest a novel approach for searching natural compounds with anti-aging and rejuvenation potential using cell cultures, with a high potential for the further in vivo applications. The present paper discusses ways of defining age for cell populations with large numbers of cells and suggests a method of assessing how young or old a cell population is based on a cell age profile approach. This approach uses experimental distributions of the cells over the cell cycle stages, acquired using flow cytometry. This paper discusses how such a profile should evolve under homeostatic maintenance of cell numbers in the proliferation niches. We describe promising results from experiments on a commercial substance claiming rejuvenating and anti-aging activity acting upon the cultures of human mononuclear cells and dermal fibroblasts. The chosen substance promotes a shift towards larger proportion of cells in synthesis and proliferation stages, and increases cell culture longevity. Further, we describe promising in vivo testing results of a selected food supplement. Based on the described concept of cell age profile and available test results, a strategy to search for natural compounds with regenerative, anti-aging and rejuvenation potential is suggested and proposed for wider and thorough testing. Proposed methodology of age assessment is rather generic and can be used for quantitative assessment of the anti-aging and rejuvenation potential of different interventions. Further research aimed at the tests of the suggested strategy using more substances and different interventions, and the thorough studies of molecular mechanisms related to the action of the substance used for testing the suggested search methodology, are needed.

## 1. Introduction

Nature is widely used as a source of active compounds for a variety of different applications, ranging from technology to food production and medicine (e.g., [1,2,3,4]). Although a large number of useful compounds is already extracted and derived from natural sources, the potential for further development is overwhelming, and in many cases can be regarded almost limitless. Many, if not majority, of natural sources used for deriving natural bioactive compounds are renewable in a relatively short time, thereby increasing the attractiveness of corresponding processes. Many natural bioactive compounds are produced at an industrial scale, but the overall impact on the environment from their applications is substantially lower than, for example, from the extraction of vegetable oils. Consequently, the nomenclature and volume of extracted bioactive compounds steadily grows reflecting growing demand (see, for example, [4,5]).

There is a large variety of holistic and therapeutic applications of natural bioactive compounds (e.g., [6,7,8,9,10,11,12]). One of the specific targets is the bio-derived compounds with anti-aging and rejuvenation potential [13,14,15,16,17,18,19,20,21,22]. Although there exists a large number of compounds claiming such activities, the issue of their assessment is extremely complicated [1,14,23,24]. It is primarily due to the variety of the effects included under different definitions of ‘anti-aging’ and ‘rejuvenation’. Scores of studies and publications are related to specific activities and specific methods of deriving bioactive compounds, but a generalized approach for their discovery and testing is yet to be developed.

This paper suggests quantifiable concepts of ‘rejuvenation’ and ‘aging’ at the cell population level. It also suggests qualitative and quantitative criteria for the assessment of rejuvenating and anti-aging activity, and formulates suggestions on searching methods for natural compounds exhibiting such properties. It also presents the results of preliminary experiments on a substance claiming such activity. According to the described age definition, the tested substance has certain rejuvenating potential, which requires further studies and independent substantiation of the effect using different methods and test protocols.

When discussing practical search strategies one should clarify the meaning of the most commonly used but rather ambiguous terms, ‘anti-aging’, ‘rejuvenation’ and ‘immortality’. They are widely used in both popular and scientific publications, but no generic definitions or generally accepted meaning when applied to humans and human life are available. Moreover, some of the distinguished authors actively protest about the widespread and non-specific use of such terms [25,26,27]. Many researchers rather prefer to speak of healthy aging and longevity rather than rejuvenation and immortality [28,29,30].

In developing a basic approach and defining anti-aging and rejuvenation activity, it is possible to start from the human as a whole, organs and functional sub-systems, or from the cell level. Indeed, a broad literature is available on the cell-level anti-aging and rejuvenation strategies, methods and studies ([31,32,33,34,35,36,37,38] can be suggested as more general publications and reviews relevant to the discussed topic). Large number of researchers directly correlate human aging to the processes at the cell level [39,40,41,42,43,44,45]. One of the explored pathways is related to cells having the highest proliferation potential [33,46,47,48] and corresponding suggestions on their applications in anti-aging and rejuvenation applications. In relation to defining if a chosen intervention has any rejuvenating effect, one should identify a measurable parameter or a group of parameters for comparing the system state ‘before’ and ‘after’ the interventions or through the changes happening with time. Within the scope of present discussion, it is also worth noting a reported link between rejuvenation and regeneration [49], and a role of the immune system in senescence and rejuvenation [50,51,52].

It was initially hypothesized that cell-based approach has a better potential for possible quantification of the aging process of live systems. Extensive experimental and theoretical studies allowed to generate and test basic concepts, and to suggest clear test protocols. Further development of the cell population age assessment approach led to the understanding of the role of proliferation niche homeostasis as a mechanism of non-pathologic cell number control. Conceptual models were further developed into computer-based simulators using cell age distribution as a specific measure of the cell system age. Extensive research was carried out with different substances claiming rejuvenating and anti-aging activity undertaken to test developed concepts, theoretical analysis and models. Collected data support the validity of the chosen approach, and it was realized that it is possible to test substances for specific activity using developed test methods and protocols.

The present paper provides a brief summary of the basic concepts of cell-based approach in quantification of the cell population aging, and of the cell number maintenance in the proliferation niche. It also describes the methodology allowing for quantified assessment of rejuvenation potential and provides an example of such assessment. This corresponding methodology can be successfully used in screening of natural compounds to select the ones with promising anti-aging and rejuvenating potential.

## 2. Results

Below we discuss the concept of age and the ways of its measurement at different levels, starting from single cells, to cell populations, tissues, organs and live organisms as a whole. This analysis is followed by the discussion on the appropriate experimental methods and test protocols. Based on the acquired results, we suggest a feasible search strategy for substances with anti-aging and rejuvenating activity. We also provide an example of testing a substance claiming anti-aging activity using the proposed methodology.

### 2.1. Anti-Aging Strategies, Rejuvenation, Immortality

#### 2.1.1. Age Assessment at the Body, Organ and Tissue Levels

There is a number of different approaches to assess human age. Chronologic age (using date of birth) is the most common example. However, it cannot take into account individual differences and it does not adequately forecast life expectancy or potential longevity. Perceived age (phenotypic approach) reflects certain individual features but is quite subjective and hard to quantify [53]. Biologic age calculated on the basis of measurable individual parameters is more adequate for the assessment of the aging process [53,54,55,56,57,58,59,60]. However, parameters and corresponding expressions used by researchers for biologic age calculation differ and, in some cases, differ quite significantly [53,54,55,56,59,60]. Values of many parameters can also vary dynamically depending on the current state of the organism, which makes application of any cumulative values including biologic age quite complicated and to a certain extent ambiguous [61,62,63,64,65,66]. Thus, although biologic age value reflects the aging process and can be used for longevity forecasts, the search for better approaches is ongoing. Interesting suggestions have been formulated in [67,68,69], where it was demonstrated that the difference between chronologic and biologic age values could be used as a good individual-level indicator of the aging process. This research was mainly focused on specific aspects related to aging, such as aging of the skin; on using chronologic and biologic age difference as possible express criteria for cosmetology [68,69]; and on the correlation between biologic age and socio-demographic and lifestyle factors in post-reproductive life [67]. However, it was also shown that this difference is correlated with age-related changes in the blood immune profile, thus reflecting deeper level changes [68,69] and can therefore better represent the aging process as compared to just biologic age.

It is clear that quantification of the aging process demands clearly defining age as a parameter for comparison. Approaches to the age definition discussed above are cumulative and use “high-level” parameters reflecting the state of human body systems, organs and tissues. For example, parameters used in the empirical expression for the biologic age calculation used in [68,69] include the forced vital lung capacity, systolic blood pressure, urea concentration in urine and blood cholesterol level. Based on the studies of different approaches to the biological age calculations one can suggest that such age definitions are derived for the “body system, organ and tissue level”, simultaneously reflecting their interrelations. The corresponding approaches can be referred to as “top-level” ones. Simultaneously, extensive research related to aging and rejuvenation is carried out at the cell level, which can be regarded as a “bottom-level” approach [32,39,40,41,42,43,44,45,46,47,48]. One of the ultimate goals in the discussed research is the desire of linking the “top” and “bottom” levels (human body-organ-tissue level as the top, and cell level as the bottom one).

Although significant progress has been achieved through both approaches, and one can correlate aging processes at “top” and “bottom” levels, a comprehensive approach unambiguously linking these two levels has not still been developed. Nevertheless, for many practical reasons the cell level seems to be more suitable for the assessment of anti-aging or rejuvenating activity of substances, especially when screening is considered. Experimenting with cell cultures is more straightforward, does not involve any prohibitive ethical questions and avoids any potential issues related to the safety of using new substances with human subjects without thorough prior investigations. Additionally, expected changes at the body-organ-tissue level may be quite slow, and using, for example, biological age as an indicator for expected rejuvenating and anti-aging effects would need a number of different tests, and ideally should involve continuous lifelong studies.

#### 2.1.2. Age Assessment for Cells and Cell Populations

Age assessment methods for cells and cell cultures seem to be more adequate from the substance screening and express testing point of view. Thus, it is important to clarify and define age as a measurable parameter in such cases. When speaking of individual cells, the concept of age is quite intuitive and straightforward: it is the time that have passed from the moment a cell was born in the process of division. However, the age definition becomes more complicated when speaking of cell populations including large numbers of individual cells, especially taking into account that continuously new cells are born and some cells are removed by apoptosis. Indeed, any cell population consists of cells with different age (as defined above), whether it is a cell culture, or the cells extracted from a live organism. Interestingly, this situation can be treated in analogy with atomic physics or chemistry: the elements at the individual cell level “are quantified” and well defined, and the situation changes when moving towards very large numbers of elements (or large “ensembles” of elements). It has quite pronounced similarity to the case of cell populations with large numbers of individual cells. Corresponding methods can be used to provide a desired link between the “bottom level” (cellular level~atomic level) and “top level” (body-organ-tissue level~continuous matter level). Physics has already developed approaches for linking the properties defined at the level of individual elements (quantification) to that at the level of large numbers of elements (ensembles, “mid-range” element numbers), and towards systems with extremely large numbers of elements (level of continuous matter). One of the formal methods providing the needed link from single element to “mid-range” element numbers is mathematical statistics.

Statistical approach is quite effective with the systems containing large numbers of similar elements, and it is intensely used in cell-related studies. For example, flow cytometry provides percentages of the cells in different states and expressing chosen biomarkers, representing results as probability values and statistical distributions, and calculating different average values. However, the statistical approach is not commonly applied to the age concept of cell population. It is clear that the “age profile” of the cell population is strictly defined only by data pool containing the age of all individual cells. From this full dataset, certain cumulative values, for example, the average age of cells, are easily calculated, but at the cost of losing certain information. In some cases, such simplification can be quite useful, but it is not fully representative and can be misleading.

According to accepted estimates, the human body contains on average about 10^14^ cells, or about 10^9^–10^10^ cells in 1 cm^3^ of the tissue. Thus, an adequate representation for the age profile of even small tissue samples needs quite a significant number of cell age values. Simplification of data representation achieved by the statistical approach is based on the modeling of the ensemble, assuming that its elements are differing in only one parameter, in our case individual cell age. It means that one can combine the elements with the same value of the parameter of interest (cell age) into sub-groups without loss of information because in such subgroups elements are otherwise indistinguishable. Plotting the numbers of cells in different age subgroups, one builds up what, in statistical analysis, is called a distribution function. Thus, one can state that the true age profile of the cell population with reasonably large numbers of cells can be adequately represented by the distribution of the cells over their individual age or the “age profile”. As is common with distribution functions, one can calculate the corresponding most probable and average age values and integrals within Quartiles. The calculations of such values are performed with a certain loss of information, and a very large number of values (age of individual cells) is than represented by very few values (average, most probable, and cumulative probabilities in intervals, such as Quartiles, etc.). In order to understand which of the “cumulative values” can better represent cell population aging, one should identify how aging and rejuvenation affect corresponding age profiles utilizing generalized distribution function formalism.

#### 2.1.3. Aging and Rejuvenation of a Cell Population as Reflected in the Age Profile

Given the above discussion, it should now be clearer why many scientists protest against using the term rejuvenation with respect to the cell level, and especially to individual cells without adequate care. It is obvious that as time cannot be reversed, the fate of a cell is predetermined: it is born through division and it functions for a certain time and ends its life through disintegration (pathological way through necrosis, or non-pathological—through apoptosis or physical cell removal). It is supposed that the process of cell division incorporates certain “error correction” mechanisms, e.g., [70], and newly divided cells in general have the same “longevity potential” and vitality as their progenitors, which is confirmed by some experimental studies, e.g., [71]. Correspondingly, as a first approximation one can state that new cells have an age equal to zero when they are born in division.

The situation is quite different for cell populations. Here, each cell is aging with time. This in turn affects the corresponding age profile, which, in itself, can dynamically change with time depending on different circumstances. It is intuitive and can be further accepted that cell populations with a larger percentage of young cells are “younger”. Changes in corresponding age profiles of the same cell population reflecting relative percentages of young and old cells before and after interventions, or simply with time, should represent the corresponding aging or rejuvenation process. With aging, the proportion of young cells should decrease and the proportion of mature and old cells should consequently increase, and vice versa. This opens the possibility for objectively assessing the anti-age activity and rejuvenating properties of substances by measuring the changes in age profiles of cell populations.

Figure 1a graphically illustrates possible changes to the cell age profile of the same cell population due to aging (corresponding profile curves are rather arbitrary and are used to visualize corresponding changes). Figure 1b illustrates the rejuvenation of the cell population reflected in its age profile due to the “addition of newly born cells” (intensified proliferation) and targeted removal of the older cells. It is also clear that because aging of different body organs and tissues can be different, e.g., [72,73], concurrent age profiles of different cell populations in the body and corresponding cell age distributions can differ even for the same subject at the same moment of time.

Simplified representations of distribution functions, using, e.g., a single average and most probable values, result in a significant loss of information (multiple values are represented by a single value). It is easy to show that with certain age profiles a significant increase in the share of young cells would not change the average age. Using the traditional definition of Quartiles and Median is somewhat better from this point of view, and can be much more practical. Formally, Quartile “cut-offs” in general case are defined as the values separating four age intervals, each containing the same number of elements (cells). Thus, for interventions with rejuvenating effects, the boundary of the first Quartile and lower Median cutoff of cell age profiles should be shifted towards smaller age values. Correspondingly, upper Median cutoff and first Quartile boundary should be shifted towards larger age values for an aging cell population.

#### 2.1.4. Aging and Rejuvenation of the Proliferation Niche

It becomes clear that the following discussion should be related to cell populations having proliferation potential, as the cells with ‘zero age’ are produced by cell division within this niche. In order to speak about cell population, rather than single cells, we should also expand the accepted concept of the **stem cell niche**. Most common concept is related to the nearest microenvironment of the stem cell [46,74,75,76,77,78,79,80,81]. Herein after, we use the term **proliferation niche** to define certain volumes containing large numbers of similar cells with active mechanisms “bringing in” newborn and “removing” old, senescent and damaged cells.

In order to discuss the aging and rejuvenation of the **proliferation niche**, one needs to specify the mechanisms of “bringing in” and “removing” the cells. As mentioned earlier, within the proposed model, “zero-aged” cells are born in the process of cell division, and this seems to be the main relevant mechanism for the majority of tissues. Of course, mass flow and diffusion (physical transport of the cells) can also take place and it is significant, for example, in the case of blood and lymph circulation. These systems are quite specific, and one can consider them as “volume-distributed” proliferation niches. There are few mechanisms leading to the “removal” of cells from a cell population. In the non-pathologic case, this occurs through apoptosis and mass transfer. Removal of outer layers of cells in mucous membranes and skin and due to tissue wounding are also examples of such ‘mass transfer’ cases. To simplify the discussion, we limit it to the cases without mass transfer.

The numbers of cells in a fixed volume of mature tissues (e.g., cell concentrations) are quite stable under non-pathologic conditions. Even the scars left after wound healing do not dramatically change the initial cell numbers. Corresponding cell number maintenance in the cell population in the presence of mechanisms, such as proliferation and apoptosis, is commonly referred to as **homeostasis** [82,83,84,85,86]. Moreover, in our case it should be applied to a proliferation niche rather than to a stem cell one. Here, selected **niche** presumes a certain defined volume with functionally and phenotypically similar cells where a maintenance of cell numbers takes place. This is a typical modeling approach with built-in approximations and simplifications. Nevertheless, such partly arbitrary definition of the proliferation niche still leads to many adequate explanations, clarifications and predictions.

If one adopts the concept of cell number maintenance (homeostasis) in the stem cell niche with its extension towards the large numbers of cells, it demands that two mechanisms leading to addition of new cells with zero age and non-pathological “removal” of the cells should be synchronized. This means that cell number maintenance in non-pathologic cases depends on the synchronization of proliferation and apoptosis that mainly targets aging, senescent, malfunctioning and damaged cells [86,87,88,89,90,91]. Thus, when averaged in time, the numbers of “added” cells should be equal to that of the “removed” ones. Such homeostatic systems should also have certain “embedded controls” monitoring the fluctuations of cell numbers around a given stable value and properly managing the correlated activities of the two mechanisms.

Without specifying the nature of “embedded control mechanisms” one can already draw certain conclusions about the possibilities for rejuvenation of homeostatic proliferation niches. Indeed, stimulating proliferation causes the numbers of young cells to increase (Figure 1b, blue sector). The cell number maintenance mechanism demands that overall cell numbers should be kept stable, and forces intensification of the apoptosis (removing predominantly older cells, indicated in Figure 1b by a gray sector). The corresponding age distribution shifts towards a larger proportion of young cells and smaller proportion of old ones, which is equivalent to the rejuvenation of the cell population. It should be noted that rejuvenation effects in systems with properly functioning cell number maintenance could be initiated by intensification of any regulating mechanism in the synchronized pair. For example, removal of the solder and senescent cells in the population forces the intensification of homeostatic proliferation, compensating for the decreasing overall cell numbers. Because the two maintenance mechanisms are synchronized, it does not matter which of the mechanisms “takes the lead”. Strategies aiming at senescent cell removal were indeed suggested, and corresponding experiments point towards the possibility of cell population rejuvenation via senescent cell clearance [92,93,94].

Now it is possible to better define the aging process using consecutive age profiles of the same cell population. Indeed, if the intensity of the cell proliferation decreases, intensity of apoptosis should also decrease, maintaining the cell numbers. As a result, on average cells will live longer, leading to large numbers of the senescent cells, which is equivalent to the cell population aging (Figure 1a). Consequently, because proliferation and apoptosis in the homeostatic proliferation niche are synchronized, aging would be caused by the fall in efficiency of these two mechanisms, independently which mechanism leads in this process. One can also conclude that interventions aimed at the prolongation of individual cell life in the homeostatic proliferation niche would not lead to rejuvenation, but to the aging of cell population. Apoptosis is forced to decrease its intensity, rarely removing senescent and old cells, in turn forcing the proliferation to decrease its intensity, thus leading to smaller proportion of young cells in the system.

Based on the **proliferation niche** concept, one can provide certain explanations for the cases when homeostasis is ‘malfunctioning’, for example, due to the failure of one of the regulating mechanisms. If for some reason apoptosis fails and at the same time proliferation is continuously active, it should lead to uncontrolled growth of cell numbers in certain local areas (neoplasms) limited only by some external factors (limited blood and nutrients supply, etc.) [88,95]. If the proliferation branch fails but apoptosis is still removing aging and damaged cells, this should consequently lead to dystrophy [96].

It should be specifically noted here that **homeostasis**, as initially defined by C.H. Waddington, refers to a **static** equilibrium. For live systems with essentially **dynamic** management (static equilibrium in live system means death), when the system follows prescribed dynamics (time-dependent pathways), he suggested the term **homeorhesis***,* or tendency of the system to follow a pre-defined generalized time trajectory [97,98]. Although the latter term is formally more appropriate, it is rarely used with living-system research (e.g., [99]), and we will use the commonly accepted term **homeostasis** keeping in mind its “dynamic extension” to living systems.

### 2.2. Screening for the Substances with Anti-Aging and Rejuvenation Potential

Based on the discussion describing cell age profile concept and homeostasis in the proliferation niche, one can formulate the basics for a method of screening substances and testing the interventions for anti-aging and rejuvenating potential using cell populations. Cell samples should be taken before a chosen intervention or substance addition and in regular intervals after that. Successful candidates should simultaneously and, in a correlated manner, increase the intensity of apoptosis and proliferation, shifting cell age profiles towards a larger proportion of younger cells. So far, the only missing link is a method that can provide corresponding cell age profiles or the information that can reflect the changes in these profiles in an adequate way. Our research has shown that good practical results can be achieved using flow cytometry and corresponding assays for studying apoptosis and cell distributions over the cell cycle stages. It is proposed that the proportion of cells in the phases immediately preceding and just following cell division adequately reflects the proportion of young cells in the population.

Experiments were carried out using representative cell cultures (in our case, human mononuclear cells and dermal fibroblasts). The corresponding cell population (culture) is split into batches (control and test ones). The distribution of the cells over cell cycle stages in both batches is measured using flow cytometry, yielding an initial age profile (control). A substance in different concentrations is added to the test batches and distributions over the cell cycle stages are measured with the cell samples taken from the cultures at regular time intervals. Substances that increase the percentage of the cells in the synthesis and proliferation stages, and in a correct way influencing the intensity of apoptosis, should have anti-aging and rejuvenation potential. Corresponding cell cultures can be either commercial laboratory lines or ex vivo ones. Chosen substances can be further assessed in vivo, for example, basing on the method suggested by Sukhovey et al. [100]. According to this suggestion, corresponding venous blood samples are taken before (control) and in regular intervals after the administration of the tested substance. Mononuclear cells are separated and tested for changes in the cell cycle stage distribution profiles and apoptosis using flow cytometry and corresponding assays. Successful substances can be further tested in pre-clinical and clinical trials, where they can be assessed using the methods accepted in geriatrics for anti-aging and rejuvenation activity at the level of the tissues, organs and the body.

Following the above discussion, it is also possible to narrow the search field. Based on the description of cell number maintenance (homeostasis) in a chosen tissue volume, one can suggest that desired substances should be naturally present in the niches where the proliferation is most intense. This idea is supported by the fact that many search strategies for anti-aging and rejuvenating substances are related to the presence of cells with high proliferation potential (stem and pluripotent animal cells) from different tissues, callus and routes from the plants and germinating seeds, etc.

### 2.3. Example of Testing Natural Substances with Claimed Anti-Aging and Rejuvenation Potential

Here, we present an example of testing naturally derived substances using the methodology suggested above. Suggested search paradigm is rather generic and does not depend on the mechanisms of action specific to certain substances and anti-aging or rejuvenation interventions. On one hand, it should allow for relatively wide application area. On the other hand, it demands thorough substantiation through independent experimental studies.

A number of commercially available substances with a claimed rejuvenation and anti-aging activity was tested. Presented results are acquired with the substance showing one of the most promising outcomes, a commercial food supplement **Ärlisätt Essential Anti-Age** [101]. Corresponding experiments were carried out with the commercial culture of human dermal fibroblasts and pooled with ex vivo culture of mononuclear cells extracted from the blood samples donated by healthy volunteers to guarantee better relevance to in vivo potential. Doses of the tested substance are given below in the μg per ml of the cell culture normalized for 1 × 10^6^ cells per mL.

Figure 2 presents the results of substance testing experiments on ex vivo human blood mononuclear cell culture. Circular diagrams represent the percentage of the cells in corresponding cell cycle stages and apoptosis acquired using flow cytometry. Columns represent corresponding profiles taken at increasing duration of cell culture incubation. The tested substance was added on Day 0 after the initial control profile was acquired. Rows represent different concentrations of the added substance (Control: no substance added; 75, 150 and 300 μg/mL). With no added substance, the culture survived for only 8 days. With the addition of 75 and 150 μg/mL on Day 0, the culture survived until Day 15 and 20, correspondingly. With the addition of 300 μg/mL on Day 0, the culture was still active until the experiment was terminated on Day 20.

Figure 3 presents the results of the substance testing experiments on human dermal fibroblast culture. Circular diagrams represent percentage of the cells in corresponding cell cycle stages and apoptosis acquired using flow cytometry. Columns represent corresponding profiles taken at increasing duration of cell culture incubation. The test substance was added on Day 0 after the initial control profile was acquired, and cell stage profiles were consecutively recorded at the end of following weeks. Rows represent different concentrations of the added substance (Control: no substance added; 70, 150 and 300 μg/mL). The experiment was terminated after 22 weeks of incubation.

Concentration effect of the tested substance was measured for both cell types. Figure 4 presents the changes in the viable cell numbers at different concentrations of added substance after 24 h of incubation as compared to the control (basic cultivation media only, no added substance). The results of the analysis after 24 h are quite representative and are provided here as a reference, however, cell viability was analyzed in all experiments. Corresponding dependences are quite characteristic. With the addition of the tested substance, number of viable cells initially decreases and starts to increase after some delay. This delay becomes smaller with increasing concentration of the added substance. In the test results after 24 h exposure, there is a drop in viable cell numbers for the concentration of 9.4 μg/mL, as the delay for the higher concentrations becomes much less than 24 h. With both cell types, increasing concentrations of added substance are causing increased intensification of the cell proliferation (up to the concentrations of 300 μg/mL). Simultaneous tests with corresponding apoptosis assays indicate that it is most probably because at higher concentrations net effect from intensification of the apoptosis exceeds that from intensification of the proliferation. This concentration was used as an initial reference. Smaller test concentrations were derived by consecutive division by 2 and higher ones by consecutive multiplication by 2, leading to the standard test protocol with the substance concentrations of 9.375; 18.75; 37.5; 75; 150; 300; 600 and 1200 μg/mL corresponding to the log_2_([C]) scale. For better clarity, the results of the experiments illustrated in Figure 2 and Figure 3 are referring to the concentrations, where the effects are most pronounced.

Estimations of half-maximal effective concentration values for the enhancement of cell proliferation referenced to the control (not taking into account decreasing values at small concentrations and concentrations exceeding 300 μg/mL) give EC50_fibroblasts_ ~62.8 and EC50_MNC_ ~125.1 μg/mL, correspondingly.

## 3. Discussion

Comparing the cell stage distribution profiles in the mononuclear cell culture experiments (Figure 2), it is clear that in treated cultures there is a significant increase in the cell proportion in synthesis and proliferation stages as compared to control (no treatment). In addition, corresponding profiles in the cultures just before they cease activity are quite characteristically dominated by the rest phase and apoptosis with a negligible percentage of cells in the synthesis and proliferation phases. It is also noticeable that the cell stage profile on Day 8 for the 75 μg/mL treated culture is very similar to that of the Control on Day 1, indicating the increased longevity of the culture. Similarly, the “pre-terminal” profile of the Control culture on Day 5 is aligned with the profile of the culture with addition of 150 μg/mL of the test substance on Day 15 (marked by red dotted arrows in Figure 2). It can also be noted that although the cell stage distribution profiles of the cultures with 150 and 300 μg/mL of added test substance visibly differ at early stages (Days 1 to 3), they become more and more similar at later incubation times. Moreover, the addition of 600 μg/mL of the test substance produced profiles not much different from that with 300 μg/mL, pointing to a certain dose–saturation effect (although the number of viable cells become smaller, see Figure 4). This effect is most pronounced at later incubation times, for example, cell stage profiles of the cultures treated with 150 and 300 μg/mL of tested substance are quite different on Day 1 but are rather similar on Days 10 and 15, correspondingly (violet dotted arrow in Figure 2).

Comparison of the cell-stage distribution profiles in the human dermal fibroblasts cell culture experiments (Figure 3) confirms general tendencies found in the experiments on mononuclear cells. Treated cultures at the end of Week 1 demonstrate a clear increase in the proportion of cells in the synthesis and proliferation stages. Although the Control culture was still viable at Week 22, it is noticeable that after Week 15 its profiles are entirely dominated by the rest phase, with a very small proportion of cells in the apoptosis, synthesis and proliferation stages. Comparing the profiles of the Control and the culture with 70 μg/mL added test substance, it is clear that similar profiles are recorded at Week 3, and at Week 15, correspondingly (blue star markers in Figure 3), indicating significant increase in culture longevity. It is also clear that the cell-stage distribution profiles of the Control at Week 1 are similar to that of 150 μg/mL-treated culture at Week 22. In addition, the profile of 150 μg/mL-treated culture at Week 22 is much “younger” than that of the Control at Week 1, having a larger percentage of the cells in the synthesis and proliferation stages. This adds at least a number of weeks of additional longevity to the treated culture beyond the timing of cited tests. In fibroblast culture experiments, we have also noticed dose–saturation effect with 600 μg/mL treatment similar to that for mononuclear cell case and decreasing numbers of the viable cells (Figure 4).

According to the developed approach, the above experiments suggest that the chosen test substance has a rejuvenating and anti-aging potential. The corresponding increase in the proportion of cells in the synthesis and proliferation cell cycle stages in the substance-treated cultures is quite clear, and is connected with increased cell culture longevity. Moreover, the changes are quite significant, and surprisingly long lasting. The experiments also indicate a certain dose–saturation effect, e.g., with the test substance concentration exceeding certain level, further changes to the profiles become rather small. It should be also noted that clear changes in the cell-stage distribution profiles with the chosen cell cultures and test substance were occurring quite quickly. For tested concentrations of the substance with both cell cultures, clear differences in the cell stage distribution profiles and apoptosis were recorded already after a few hours after treatment. This also suggests that the proposed method has a good potential for prompt substance screening.

Experimental data support the suggestion that age profile in cell population can be adequately represented by the distribution of the cells over cell cycle stages. Moreover, corresponding anti-aging and rejuvenating potential of the interventions can be directly correlated with the changes in cell-stage distribution profiles. Additional rationale for using results of flow cytometry comes from cell cycle stage measurements [102,103,104,105,106]. Studies on the duration of corresponding cell cycle stages for functioning cells indicate that the longest stage in the cell life cycle is the rest phase, and the transiting synthesis and proliferation phases are relatively fast [104,105,106]. Thus, the proportion of cells in the synthesis and proliferation phases should be directly correlated to the numbers of newly born cells in the chosen population.

In support of the choice of the cell cultures used for testing, one can return to the discussion on the mechanisms of “bringing in” new cells and “removing” senescent ones. As pointed out earlier, along with the intrinsic mechanisms (proliferation and apoptosis) one should also consider mass flow, which makes hematopoietic systems quite different from many others. Moreover, hematopoietic system is generally ‘more dynamic’ in response to various extrinsic factors and interventions, while dermal one can be considered much more conservative in terms of changes. Detected similarities in the effect caused by the tested interventions upon the cultures of cells belonging to such different systems give more credibility to the results and conclusions. There is also an argument in favor of using the proportion of cells in the cell cycle stages just preceding, and immediately following division as a basic criterion rather than targeting the numbers of stem and progenitor cells, preferring indicators of direct action to the ones reflecting the proliferation potential.

Direct experiments in support of the discussed concepts and methodology were also performed in vivo. Moreover, following the above discussions, corresponding substances should also have good tissue regeneration potential. Preliminary in vivo studies with healing of skin lesions, burns and cuts confirm this suggestion. Experiments were also performed on the mononuclear cells extracted from the blood donated by healthy volunteers before and after taking the described food supplement. In all cases, it was possible to detect changes in the cell cycle stage profile towards an increasing proportion of cells in the synthesis and proliferation phases starting after 30–40 min, and in some cases detectable up to 20–30 days later. Although it does not confirm the rejuvenating effect of the particular food supplement without any doubts, it points to a tangible anti-aging and rejuvenation potential and it definitely deserves further in-depth studies.

It should be specifically noted that presented material is focusing on describing suggested generic search paradigm and methodology allowing for quantitative assessment of anti-aging and rejuvenation potential of different substances and interventions. Because of that it has a good potential for wide applications. Simultaneously, suggested methodology demands thorough independent substantiation. It is also clear that natural substance chosen just as an example of search strategy application deserves much more attention. Although its properties were studied, certain suggestions on the mechanisms of its action are proposed and corresponding conceptual and computer models are generated and to certain extent experimentally substantiated, more detailed studies of molecular mechanisms are needed. Analysis of these findings will be presented in separate publications, and in turn would need independent substantiation.

## 4. Materials and Methods

### 4.1. Extraction and Cultivation of Human Blood Mononuclear Cells

To increase the relevance of testing for the living organism, corresponding mononuclear cells were extracted from material donated by 26 healthy volunteers aged 18 to 50 years (11 male and 15 female, mean age 31.6 ± 10.1 years). Extraction of mononuclear cells from the venous blood was performed in a medical laboratory environment using the protocol generally accepted in microbiology [107]. Venous blood is taken from the ulnar vein in the morning before any meals. The separation of mononuclear cells from the blood is performed by gradient centrifugation using DIACOLL-1077 (density 1.077, gradient cell separation medium). The heparinized blood is diluted 3-fold with a Versene solution (an ethylenediaminetetraacetic acid solution, and a non-enzymatic cell dissociation reagent), then layered on top of the DIACOLL-1077 density gradient and centrifuged at 300× *g* for 45 min. The interphase containing mononuclear cells is collected and washed three times with a cell culture medium RPMI-1640. Mononuclear cells were diluted to the normalized cell concentration of 1 × 10^6^ mL^−1^. Then, 5 mL samples were placed into standard 24-well incubation racks using Gibco Cell Culture Media RPMI 1640 containing 10% of thermally inactivated bovine calf serum (BioClot GmbH, Aidenbach, Germany), and were cultivated in the CO_2_ incubator by Sanyo (5% CO_2_, 37 °C, and 95% RH, Osaka, Japan) up to 23 days with or without test substance. See Figure 2 for the corresponding incubation duration of the samples without (control) and with the added test substance. Corresponding doses of the tested substance are given in μg per mL of the normalized cell culture with 1 × 10^6^ cells/mL. Immuno-phenotyping of cultivated cells was carried out in regular intervals using a commercial flow cytometer CytoFLEX (Beckman Coulter Inc., Brea, CA, USA).

### 4.2. Cultivation of the Human Fibroblasts

Healthy primary dermal fibroblast cells (line Hs27) were purchased from Merck (Rahway, NJ, USA) and cultivated together with activated human mononuclear cells in the Dulbecco’s Modified Eagle cell cultivation Medium DMEM/F-12 by Gibco (Waltham, MA, USA), with addition of 2 mMol/L glutamine and 10% of thermally inactivated bovine calf serum (BioClot GmbH). Cultures were seeded into Petri dishes at a density of 2 × 10^5^ cells/cm^2^ and placed into the CO_2_ incubator by Sanyo. Incubation was carried out at standard conditions (5% CO_2_, 37 °C, and 95% relative humidity) up to 23 weeks with or without test substance. See Figure 3 for the corresponding incubation duration of the samples without (control) and with the added test substance. Corresponding doses of the tested substance are given in μg per ml of the normalized cell culture with 1 × 10^6^ cells/mL. Immuno-phenotyping of cultivated cells was carried out in regular intervals using a commercial flow cytometer.

### 4.3. Flow Cytometry Analysis

The following procedure was used for the flow cytometry analysis of the cell cycle stage distribution profile. Cell samples were taken from the culture, re-suspended in the isotonic solution (0.9% NaCl) and filtered (Filicons, by Becton Dickinson Biosciences, Franklin Lakes, NJ, USA, pore diameter 50 μ). Cell concentration in the resulting sample was normalized to 1 × 10^6^ mL^−1^. Corresponding doses of the tested substance are given in μg per ml of the cell culture normalized for 1 × 10^6^ mL^−1^.

Analysis was carried out using a CytoFLEX system (Beckman Coulter Inc., Brea, CA, USA), monitoring specific markers using cell cycle stage and apoptosis analysis assays by Beckman Coulter Inc., Brea, CA, USA. Prior to the analysis cells were re-suspended in the phosphate-buffered saline (PBS). During each test, 50,000 to 75,000 cells were analyzed. Results of the tests were analyzed and saved using the firmware of the flow cytometry machine.

### 4.4. Cell Viability Analysis

Viability of the cells was controlled for all experiments cited above. Fibroblast viability was determined by colorimetric analysis using 3-(4,5-dimethylthiazol-2-yl)-2,5-diphenyltetrazolium bromide (MTT) by Atocel, Graz, Austria. Cells were seeded in a 96-well microplate at the initial concentration of 5 × 10^3^ mL^−1^ and treated with Dulbecco’s Modified Eagle Medium (DMEM) in the groups with the added tested substance at different concentrations and a control (only DMEM and no substance). The medium was changed every 3 days. For the measurements, the medium was replaced with a 15% solution of MTT in phosphate-buffered saline (5 mg/mL) and incubated at 37 °C for 2 h. After that, the solution was replaced with dimethylsulfoxide and shaken in a dark chamber for 15 min. Optical density for each well was measured using EnVision Multilabel Plate Reader (PerkinElmer, Waltham, MA, USA) at a wavelength of 570 nm. The number of cells was determined through the optical density readout according to the calibration curves. The tests were repeated six times for each sample. Mononuclear cell viability was determined using Trypan Blue Exclusion Test according to standard protocol (e.g., [108]). Unstained cells were counted using an inverted microscope CKX-41 (Olympus, Shinjuku City, Japan). Each experiment was reproduced 6 times.

## 5. Conclusions

The proposed strategy for searching for natural compounds with anti-aging and rejuvenation potential in this study is based on the concepts of defining cell population age and homeostasis in the proliferation niche. Suggested test procedures include flow cytometry analysis (distribution of the cells over cell cycle stages, and apoptosis). Such cell-centered strategy has certain advantages over in vivo studies: they are faster and free of many ethical and safety issues. Corresponding test protocol is using assessment of the relative share of young and old cells in cell population, and intensity of the apoptosis, and monitoring the changes in corresponding statistical distributions acquired via flow cytometry experiments. Proportion of cells in the phases of cell cycles directly preceding and immediately following cell division measured using flow cytometry is chosen as a main parameter used for comparison. Cell culture experiments carried out with a number of candidates, including a commercial substance claiming anti-aging and rejuvenating activity, confirm that a shift towards increased proportion of cells in the synthesis and proliferation phases is directly linked to increased longevity of the cell cultures. Data acquired through in vitro tests with selected substance presented in this paper support the presence of anti-aging and rejuvenation potential.

Initial in vivo experiments confirmed that the tested substance has a distinct rejuvenating potential according to the chosen criteria. The changes in the blood mononuclear cell age profile of volunteers showed a clear shift towards a larger proportion of the cells in the synthesis and proliferation stages as detected shortly after using the test substance, and this shift was detectable for a substantial amount of time. Preliminary in vivo experiments also demonstrated that chosen substance used together with hyaluronate carrier is also capable of improving tissue regeneration.

Suggested search paradigm is rather generic and does not depend on the mechanisms of action specific to certain substances and anti-aging or rejuvenation interventions. On one hand, it should allow for relatively wide application areas. On the other hand, it requires thorough substantiation through independent experimental studies. Natural substance chosen just as an example of search strategy application also deserves much more attention. Its main properties are studied, suggestions on the mechanisms of its action are proposed and corresponding conceptual and computer models are generated and to a certain extent experimentally substantiated. However, scope of the present publication does not allow for the detailed description of studies related to the molecular mechanisms of its action. Analysis of these findings will be presented in separate publications, and in turn would need independent substantiation.

## Figures and Tables

**Figure 1 ijms-24-08020-f001:**
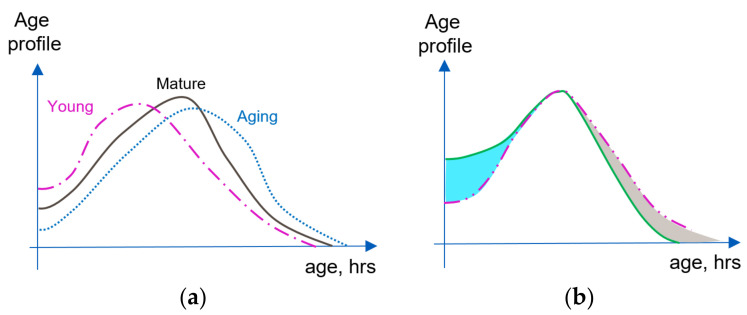
Comparing age profiles of a cell population. (**a**) Age profile changes corresponding to the changes from young to mature and old cell populations. (**b**) Changes in the age profile of cell population leading to rejuvenation: result of intensifying proliferation (blue section); result of older cell removal for example due to the intensification of apoptosis (grey section). Dotted line: original population age profile; solid green line: result of two consecutive rejuvenating interventions.

**Figure 2 ijms-24-08020-f002:**
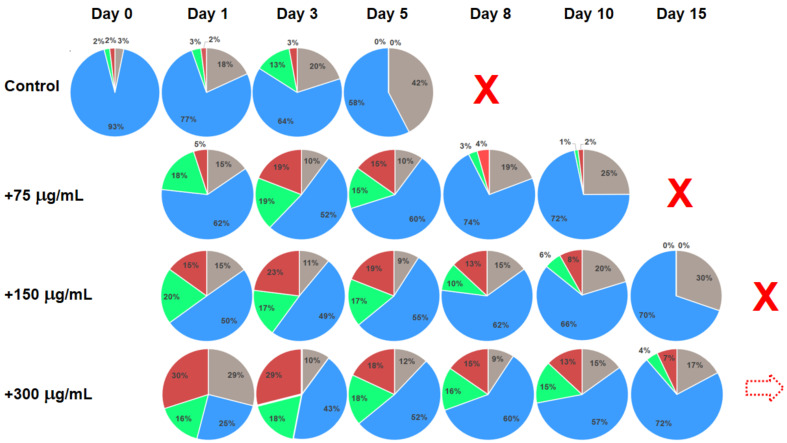
Changes in the cell cycle stage distributions for the ex vivo human mononuclear cell culture with different concentrations of the added substance represented by circular diagrams. Test substance was added on Day 0 after the initial cell cycle stage distribution was acquired. Control: no added substance. Grey sectors: apoptosis; blue sectors: rest phase (G0 + G1); green sectors: synthesis; red sectors: proliferation (G2 + M). Dotted arrows indicate similar profiles of cell cycle stage distributions. Red cross symbols identify that corresponding cultures expired and the experiment was stopped. Dotted red arrow symbol in the last identifies that corresponding culture was still active after day 15.

**Figure 3 ijms-24-08020-f003:**
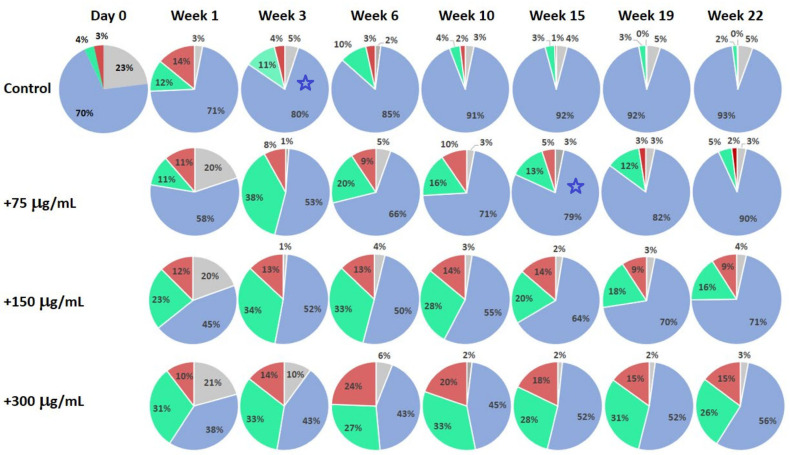
Changes in the cell cycle stage distributions for human dermal fibroblast culture with different concentrations of the added test substance represented by circular diagrams. The substance was added on Day 0 after initial cell cycle stage distribution was acquired. Control: no added substance. Grey sectors: apoptosis; blue sectors: rest phase (G0 + G1); green sectors: synthesis; red sectors: proliferation (G2 + M). Blue stars indicate closely matching profiles.

**Figure 4 ijms-24-08020-f004:**
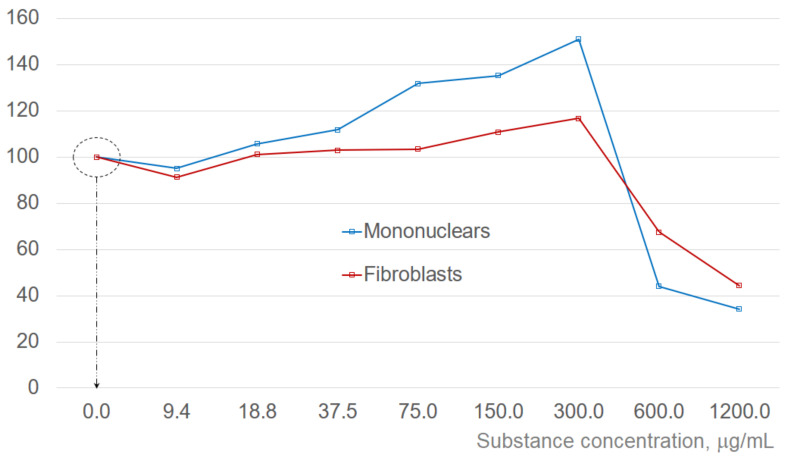
Changes in the viable cell numbers 24 h after the addition of tested substance in different concentrations. Corresponding values are shown in percent, normalized to the control (cell numbers in the culture containing only basic incubation media). Dotted circle indicates the cell number in the control (100%). Substance concentration scale is logarithmic. Corresponding dotted arrow and dotted circle identify the initial condition (viable cell numbers before the substance is added).

## Data Availability

Corresponding data are handled according to the EU regulations and GDPR & ePrivacy Directive. Basic data are available on request due to ethical and personal data handling restrictions.

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
