# Peer review of "Novel Strategy in Searching for Natural Compounds with Anti-Aging and Rejuvenating Potential"

_ijms, 2023, doi:10.3390/ijms24098020_

Round 1

Reviewer 1 Report

The present research article by Koptyug et al. entitled “Novel strategy in searching for natural compounds with anti-aging and rejuvenating potential” demonstrates the different approaches of defining age for cell populations with large numbers of cells and provides a method of assessing aging process, based on a cell-age profile approach. Authors used experimental distributions of the cells over the cell cycle stages, acquired by flow cytometry. In addition, authors showed how such a profile should evolve under homeostatic maintenance of cell numbers in the proliferation niches. There are few queries that need to be addressed as listed below:

Major queries/comments:

#1. In Fig.1a and 1b, authors compare the age profiles of a cell population graphically i.e. x- & y coordinates, however no units e.g. numbers/time/days/weeks are mentioned in the graph. Is there any specific reason? It would be better to represent the correlation between aging and cell population by using some measurable units.

#2. In Fig. 2, authors mentioned the different concentrations (75 ug/ml, 150 ug/ml and 300 ug/ml) of added substances to the ex vivo human mononuclear cell culture. How did authors choose those concentrations? Authors need to show a dose response curve for EC50 values of each substance used in the study. Also, which method was used to evaluate viability/toxicity of the substances added to the cultured cells? What percent of cells were viable upon adding the substances.

#3. In Fig. 3, I have similar concern as raised in query#2, authors mentioned the different concentrations (75 ug/ml, 150 ug/ml and 300 ug/ml) of added substances to the cultured human dermal fibroblast culture. How did authors choose those concentrations? Authors need to show a dose response curve for EC50 values of each substance used in the study. Also, which method was used to evaluate viability/toxicity of the substances added to the cultured cells? What percent of cells were viable upon adding the substances.

#4. Did author use any additional control cell line e.g. immortalized HEK293/He La cells? Authors should include an additional control cell line th

Author Response

Response to Review 1

We are grateful to the reviewer for the interest to our studies, thorough attention to the text and content of our manuscript and useful comments and suggestions. We accept all suggestions, although it is hard to specifically reply to the general comments indicated by crosses (x), so please find the replies below in the part ‘Answering to general comments’.

Specific comments are answered position by position further below.

Answering to general comments

(i) Manuscript was proof read by a native English speaking person after spelling and grammar check performed under MsWord.

(ii) Special attention was focused at better clarifying the text related to the approach, protocols and explanations.

(iii) Additional paragraphs are added to the Introduction:
“ It was initially hypothesized that cell-based approach has a better potential for possible quantification of the aging process of live systems. Extensive experimental and theoretical studies allowed to generate and test basic concepts, and to suggest clear test proto-cols. Further development of the cell population age assessment approach led to the understanding of the role of proliferation niche homeostasis as a mechanism of non-pathologic cell number control. Conceptual models were further developed into computer-based simulators using cell age distribution as a specific measure of the cell system age. Extensive research was carried out with different substances claiming rejuvenating and antiaging anti-aging activity undertaken to test developed concepts, theoretical analysis and models. Collected data support the validity of chosen approach, and it was realized that it is possible to test substances for specific activity using developed test methods and protocols.

Present paper provides a brief summary of the basic concepts of cell-based approach in quantification of the cell population aging, and of the cell number maintenance in the proliferation niche. It also describes the methodology allowing for quantified assessment of rejuvenation potential and provides an example of such assessment. Corresponding methodology can be successfully used in screening of natural compounds selecting the ones with promising antiaging anti-aging and rejuvenating potential.”

(iv) Abstract and Conclusions are updated to better reflect the content of the paper. Now the text for the Conclusions is formulated as follows:
“Proposed strategy for searching for natural compounds with anti-aging and rejuvenation potential is basing on the concepts of defining cell population age and homeostasis in the proliferation niche. Suggested test procedures are using flow cytometry analysis (distribution of the cells over cell cycle stages, and apoptosis). Such cell-centered strategy has certain advantages over in vivo studies, being faster and free of many ethical and safety issues. Corresponding test protocol is using assessment of the relative share of young and old cells in cell population, and intensity of the apoptosis, and monitoring the changes in corresponding statistical distributions acquired via flow cytometry experiments. Proportion of cells in the phases of cell cycles directly preceding and immediately following cell division measured by flow cytometry is chosen as a main parameter used for comparison. Cell culture experiments carried out with a number of candidates, including a commercial substance claiming anti-aging and rejuvenating activity, confirm that a shift towards increased proportion of cells in the synthesis and proliferation phases is directly linked to increased longevity of the cell cultures. Data acquired through in vitro tests with selected substance presented in the paper support the presence of anti-aging and rejuvenation potential. 

Initial in vivo experiments confirmed that tested substance has a distinct rejuvenating potential according to the chosen criteria. The changes in the blood mononuclear cell age profile of volunteers showed a clear shift towards a larger proportion of the cells in the synthesis and proliferation stages as detected shortly after using the test substance, and this shift was detectable for a substantial amount of time. Preliminary in vivo experiments also demonstrated that chosen substance used together with hyaluronate carrier is also capable of improving tissue regeneration.

Based on the above results, the proposed strategy is suggested for wider independent testing and future applications. The authors are continuing both basic research for providing better understanding of the underlying mechanisms, and are testing a number of other substances for possible applications.  “

Answering to specific comments

Reviewer 1

(R1.1) “#1. In Fig.1a and 1b, authors compare the age profiles of a cell population graphically i.e. x- & y coordinates, however no units e.g. numbers/time/days/weeks are mentioned in the graph. Is there any specific reason? It would be better to represent the correlation between aging and cell population by using some measurable units.”

Figure 1 is modified and the time units are added

(R1.2) “#2. In Fig. 2, authors mentioned the different concentrations (75 ug/ml, 150 ug/ml and 300 ug/ml) of added substances to the ex vivo human mononuclear cell culture. How did authors choose those concentrations? Authors need to show a dose response curve for EC50 values of each substance used in the study. …”

Additional text and Figure 4 are added into the text before ‘2. Discussion’:

“Concentration effect of the tested substance was measured for both cell types. Figure 4 presents the changes in the viable cell numbers at different concentrations of added sub-stance after 24 hours of incubation as compared to the control (basic cultivation media only, no added substance). The results of analysis after 24hrs are quite representative and are provided here as a reference, however cell viability was analyzed in all experiments. Corresponding dependences are quite characteristic. With the addition of tested substance, number of viable cells is initially decreasing and starts to increase after some delay. This delay becomes smaller with increasing con-centration of added substance. In the test results after 24-hour exposure it is showing as a drop in viable cell numbers for the concentration of 9.4 mkg/ml, as the delay for the higher concentrations is becoming much less than 24 hours. both cell types, increasing concentrations of added substance are causing increasing intensification of the cell proliferation (up to the concentrations about of 300 mkg/ml). Simultaneous tests with corresponding apoptosis analysis assays indicate, that it is most probably due to the fact because at higher concentrations at the concentrations above 300 mkg/ml net effect from intensification of the apoptosis exceeds the that capacity of enhanced from intensification of the proliferation. This concentration was used as an initial reference. Smaller test concentrations were derived by consecutive division by 2 and higher ones- by consecutive multiplication by 2 leading to the standard test protocol with the substance concentrations of 9.375; 18.75; 37.5; 75; 150; 300; 600 and 1200 mkg/ml corresponding to the log2([C]) scale. For better clarity, the results of the experiments illustrated in Figures 2 and 3 are referring to the concentrations, where the effects are most pronounced.         

 (Figuren - in attached word document)

Figure 4. Changes in the viable cell numbers 24 hours after the addition of tested substance in different concentrations. Corresponding values are shown in percent, normalized to the control (cell numbers in the culture containing only basic incubation media). Dotted circle indicates the cell number in the control (100%). Substance concentration scale is logarithmic.

Estimations of half-maximal effective concentration values for the enhancement of cell proliferation referenced to the control (not taking into account decreasing values at small concentrations and concentrations exceeding 300 mkg/ml) give EC50fibroblasts ~ 62.8 and EC50MNC ~ 125.1 mkg/ml correspondingly.”

(R1.3) “#2. … Also, which method was used to evaluate viability/toxicity of the substances added to the cultured cells? What percent of cells were viable upon adding the substances.

 New section is added:
“4.4. Cell viability analysis

Viability of the cells was controlled for all experiments cited above. Fibroblast viabil-ity was determined by colorimetric analysis using 3-(4,5-dimethylthiazol-2-yl)-2,5-diphenyltetrazolium bromide (MTT) by Atocel, Austria. Cells were seeded in a 96-well microplate at the initial concentration of 5×103 ml-1 and treated with Dulbecco's Modified Eagle Medium (DMEM) in the groups with added tested substance at different concentrations and a control (only DMEM and no substance). The medium was changed every 3 days. For the measurements, the medium was replaced with a 15% solution of MTT in phosphate-buffered saline (5 mg/ml) and incubated at 37°C for 2 hours. After that, solution in a well was replaced with dimethylsulfoxide and shaken in a dark chamber for 15 minutes. Optical density for each well was measured using En-Vision Multilabel Plate Reader (PerkinElmer, Waltham, MA, USA) at a wavelength of 570 nm. The number of cells was determined through the optical density readout according to the calibration curves. The tests were repeated six times for each sample. Mononuclear cell viability was determined using Trypan Blue Exclusion Test according to standard proto-col (e.g. [108]). Unstained cells were counted using an inverted microscope CKX-41 (Olympus, Japan). Each experiment was reproduced 6 times.”

Andrey V. Koptyug

                                          March 31st, Östersund, Sweden

Reviewer 2 Report

This study investigated the strategies in searching for antiaging natural compounds. The results of this method are promising but lack molecular mechanisms. The article is well organized, and its presentation is good and have found interesting strategies for natural antiaging compounds.

The overall article represents preliminary data and needs to investigate detailed molecular mechanisms including telomere and epigenetics analysis.

Author Response

Response to Review 2

We are grateful to the reviewer for the interest to our studies, thorough attention to the text and content of our manuscript and useful comments and suggestions. We accept all suggestions, although it is hard to specifically reply to the general comments indicated by crosses (x), so please find the replies below in the part ‘Answering to general comments’.

Specific comments are answered position by position further below.

Answering to general comments

(i) Manuscript was proof read by a native English speaking person after spelling and grammar check performed under MsWord.

(ii) Special attention was focused at better clarifying the text related to the approach, protocols and explanations.

(iii) Additional paragraphs are added to the Introduction:
“ It was initially hypothesized that cell-based approach has a better potential for possible quantification of the aging process of live systems. Extensive experimental and theoretical studies allowed to generate and test basic concepts, and to suggest clear test proto-cols. Further development of the cell population age assessment approach led to the understanding of the role of proliferation niche homeostasis as a mechanism of non-pathologic cell number control. Conceptual models were further developed into computer-based simulators using cell age distribution as a specific measure of the cell system age. Extensive research was carried out with different substances claiming rejuvenating and antiaging anti-aging activity undertaken to test developed concepts, theoretical analysis and models. Collected data support the validity of chosen approach, and it was realized that it is possible to test substances for specific activity using developed test methods and protocols.

Present paper provides a brief summary of the basic concepts of cell-based approach in quantification of the cell population aging, and of the cell number maintenance in the proliferation niche. It also describes the methodology allowing for quantified assessment of rejuvenation potential and provides an example of such assessment. Corresponding methodology can be successfully used in screening of natural compounds selecting the ones with promising antiaging anti-aging and rejuvenating potential.”

(iv) Abstract and Conclusions are updated to better reflect the content of the paper. Now the text for the Conclusions is formulated as follows:
“Proposed strategy for searching for natural compounds with anti-aging and rejuvenation potential is basing on the concepts of defining cell population age and homeostasis in the proliferation niche. Suggested test procedures are using flow cytometry analysis (distribution of the cells over cell cycle stages, and apoptosis). Such cell-centered strategy has certain advantages over in vivo studies, being faster and free of many ethical and safety issues. Corresponding test protocol is using assessment of the relative share of young and old cells in cell population, and intensity of the apoptosis, and monitoring the changes in corresponding statistical distributions acquired via flow cytometry experiments. Proportion of cells in the phases of cell cycles directly preceding and immediately following cell division measured by flow cytometry is chosen as a main parameter used for comparison. Cell culture experiments carried out with a number of candidates, including a commercial substance claiming anti-aging and rejuvenating activity, confirm that a shift towards increased proportion of cells in the synthesis and proliferation phases is directly linked to increased longevity of the cell cultures. Data acquired through in vitro tests with selected substance presented in the paper support the presence of anti-aging and rejuvenation potential. 

Initial in vivo experiments confirmed that tested substance has a distinct rejuvenating potential according to the chosen criteria. The changes in the blood mononuclear cell age profile of volunteers showed a clear shift towards a larger proportion of the cells in the synthesis and proliferation stages as detected shortly after using the test substance, and this shift was detectable for a substantial amount of time. Preliminary in vivo experiments also demonstrated that chosen substance used together with hyaluronate carrier is also capable of improving tissue regeneration.

Based on the above results, the proposed strategy is suggested for wider independent testing and future applications. The authors are continuing both basic research for providing better understanding of the underlying mechanisms, and are testing a number of other substances for possible applications.  “

Answering to specific comments

 (R2.1) “The overall article represents preliminary data and needs to investigate detailed molecular mechanisms including telomere and epigenetics analysis.”

We are grateful to the reviewer for this comment.

Indeed, these are the results of certain preliminary studies, mainly focusing at the ‘global level’ of effects. Actually, results of many experiments and investigations held during eight-year project are not included into this paper. We already feel that the planned focus of the paper towards the strategy of searching for substances with anti-aging and rejuvenating potential is to some extent overwhelmed by very promising results for a particular substance used just as an example. We are continuing the studies into the details of the molecular mechanisms. Some of the most ‘feasible’ hypotheses we have formulated are unfortunately proven unsubstantiated. More suggestions are formulated and many additional experiments are under the way and planned.

Andrey V. Koptyug

                                          March 31st, Östersund, Sweden

Author Response

(The authors gave the same response as above.)

Round 2

Reviewer 1 Report

Revised version of the manuscript has been improved substantially and addressed all queries/comments. 

Author Response

Many thanks for your efforts. 

Through the comments from two other reviewers we have realized that main focus of their attention was not to the suggested search strategy, but to the substance used only nas an example of its application. In the updated version of manuscript additional text was added to stress the main aim of the paper. 

Reviewer 2 Report

I understand the molecular mechanism experiments are continued but I feel it is not provided in this manuscript and lack supporting data. 

Author Response

We are thankful to the reviewer for a high interest to the particular substance used as an example.

We have performed specific experiments aiming at studying its properties and molecular mechanisms related to its action. Moreover, this substance was not only extracted but also analytically purified in the amounts adequate for extensive studies.

    However, the aim of current paper is to attract attention from specialists to a potential search methodology, and cited experiments with particular substance are used to support our suggestions. Correspondingly, the focus in present paper is given to defining the age of cell population, to the search methodology for the substances allowing good quantification of the results and not to the particular substance, its properties and corresponding mechanisms of its action. We believe that used example gives certain credibility to the potential of search methodology, as it is rather generic and does not depend on the properties of specific substances. Indeed, studies related to the chosen substance are of utmost importance. There are significant results in both modeling and experimental studies of corresponding mechanisms. However, substantiation of the models and mechanisms demands significantly more journal space than a single publication can accommodate. At least two more publications are already in the pipeline, and a 52-page pct patent application is in the hands of Swedish national patent office since November 2022.

    Saying this, we recognize that the main aim of present publication should be outlined much clearer, and corresponding alterations should be incorporated into the abstract, main text and conclusions.

Incorporated changes

Abstract, first few lines (now lines 10 to 13) are changed to:

“We suggest a novel approach of searching for natural compounds with anti-aging and rejuvena-tion potential using cell cultures, with a high potential for the further in vivo applications. Pre-sent paper discusses ways of defining age for cell populations with large numbers of cells and suggests a method of assessing how young or old a cell population is basing on a cell age profile approach. “

Abstract, last lines (now lines 20 to 27) are changed to:

“Based on the described concept of cell age profile and available test results a strategy to search for natural compounds with regenerative, anti-aging and rejuvenation potential is suggested and proposed for wider and thorough testing. Proposed methodology of age assessment is rather generic and can be used for quantitative assessment of the anti-aging and rejuvenation potential of different interventions. Further research is aiming at the tests of suggested strategy using more substances and different interventions, and at thorough studies of molecular mechanisms related to the action of the substance used for testing of suggested search methodology.”

Keywords: added “search strategy”

Introduction, current lines 55 to 57, changed to:

“According to described age definition, tested substance has certain rejuvenating potential, which deserves further studies and independent substantiation of the effect using different methods and test protocols. “  

2.3. Example of testing natural substances with claimed anti-aging and rejuvenation potential, first lines (now lines 380-385) are changed to:

“Here we present an example of testing naturally derived substances using the methodology suggested above. Suggested search paradigm is rather generic and does not de-pend on the mechanisms of action specific to certain substances and anti-aging or rejuvenation interventions. On one hand, it should allow for relatively wide application area. On the other hand, it demands thorough substantiation through independent experimental studies. “

At the end of ‘Results’ following paragraph is added (now lines 545-555):

“It should be specifically noted, that presented material is focusing at describing suggested generic search paradigm and methodology allowing for quantitative assessment of anti-aging and rejuvenation potential of different substances and interventions. Because of that, it has a good potential within wide application area. Simultaneously, suggested methodology demands thorough independent substantiation. It is also clear that natural substance chosen just as an example of search strategy application deserves much more attention. Although its properties were studied, certain suggestions on the mechanisms of its action are proposed and corresponding conceptual and computer models are generated and to certain extent experimentally substantiated, more detailed studies of molecular mechanisms are needed. Analysis of these findings will be presented in separate publications, and in turn would need independent substantiation.  “

Following paragraph is added to the ‘Conclusions’ (now lines 649-659):

“Suggested search paradigm is rather generic and does not depend on the mechanisms of action specific to certain substances and anti-aging or rejuvenation interventions. On one hand, it should allow for relatively wide application area. On the other hand, it requires thorough substantiation through independent experimental studies. Natural substance chosen just as an example of search strategy application also deserves much more attention. Its main properties were studied, suggestions on the mechanisms of its action are proposed and corresponding conceptual and computer models are generated and to certain extent experimentally substantiated. However, scope of present publication does not allow for the detailed description of studies related to the molecular mechanisms of its action. Analysis of these findings will be presented in separate publications, and in turn would need independent substantiation. “

Reviewer 3 Report

When studying anti-ageing and rejuvenating properties flow cytometry is on approach, but it is also possible to use other marker-based analysis, such as activity or levels of RNA or protein-based markers. I would like to see a correlations and discussions related to studies where marker based on protein, RNA or DNA were used to study these processes and how these are correlated to flow cytometry based studies. Please refer to this type of studies in introduction and discussion.

Author Response

(The authors gave the same response as above.)

Round 3

Reviewer 2 Report

I agree with the author's explanation.